# The Shared Proteome of the Apomictic Fern *Dryopteris affinis* ssp. *affinis* and Its Sexual Relative *Dryopteris oreades*

**DOI:** 10.3390/ijms232214027

**Published:** 2022-11-14

**Authors:** Sara Ojosnegros, José Manuel Alvarez, Jonas Grossmann, Valeria Gagliardini, Luis G. Quintanilla, Ueli Grossniklaus, Helena Fernández

**Affiliations:** 1Area of Plant Physiology, Department of Organisms and Systems Biology, University of Oviedo, 33071 Oviedo, Spain; 2Functional Genomic Center Zurich, University and ETH Zurich, 8092 Zurich, Switzerland; 3SIB Swiss Institute of Bioinformatics, 1015 Lausanne, Switzerland; 4Department of Plant and Microbial Biology & Zurich-Basel Plant Science Center, University of Zurich, 8006 Zurich, Switzerland; 5Department of Biology and Geology, Physics and Inorganic Chemistry, University Rey Juan Carlos, 28933 Móstoles, Spain

**Keywords:** apogamy, *Dryopteris affinis* ssp. *affinis*, *Dryopteris oreades*, fern, plant defense, plant stress, proteomics, STRING database

## Abstract

Ferns are a diverse evolutionary lineage, sister to the seed plants, which is of great ecological importance and has a high biotechnological potential. Fern gametophytes represent one of the simplest autotrophic, multicellular plant forms and show several experimental advantages, including a simple and space-efficient in vitro culture system. However, the molecular basis of fern growth and development has hardly been studied. Here, we report on a proteomic study that identified 417 proteins shared by gametophytes of the apogamous fern *Dryopteris affinis* ssp. *affinis* and its sexual relative *Dryopteris oreades*. Most proteins are predicted to localize to the cytoplasm, the chloroplast, or the nucleus, and are linked to enzymatic, binding, and structural activities. A subset of 145 proteins are involved in growth, reproduction, phytohormone signaling and biosynthesis, and gene expression, including homologs of SHEPHERD (SHD), HEAT SHOCK PROTEIN 90-5 (CR88), TRP4, BOBBER 1 (BOB1), FLAVONE 3’-O-METHYLTRANSFERASE 1 (OMT1), ZEAXANTHIN EPOXIDASE (ABA1), GLUTAMATE DESCARBOXYLASE 1 (GAD), and dsRNA-BINDING DOMAIN-LIKE SUPERFAMILY PROTEIN (HLY1). Nearly 25% of the annotated proteins are associated with responses to biotic and abiotic stimuli. As for biotic stress, the proteins PROTEIN SGT1 HOMOLOG B (SGT1B), SUPPRESSOR OF SA INSENSITIVE2 (SSI2), PHOSPHOLIPASE D ALPHA 1 (PLDALPHA1), SERINE/THREONINE-PROTEIN KINASE SRK2E (OST1), ACYL CARRIER PROTEIN 4 (ACP4), and NONHOST RESISTANCE TO P. S. PHASEOLICOLA1 (GLPK) are worth mentioning. Regarding abiotic stimuli, we found proteins associated with oxidative stress: SUPEROXIDE DISMUTASE[CU-ZN] 1 (CSD1), and GLUTATHIONE S-TRANSFERASE U19 (GSTU19), light intensity SERINE HYDROXYMETHYLTRANSFERASE 1 (SHM1) and UBIQUITIN-CONJUGATING ENZYME E2 35 (UBC35), salt and heavy metal stress included MITOCHONDRIAL PHOSPHATE CARRIER PROTEIN 3 (PHT3;1), as well as drought and thermotolerance: LEA7, DEAD-BOX ATP-DEPENDENT RNA HELICASE 38 (LOS4), and abundant heat-shock proteins and other chaperones. In addition, we identified interactomes using the STRING platform, revealing protein–protein associations obtained from co-expression, co-occurrence, text mining, homology, databases, and experimental datasets. By focusing on ferns, this proteomic study increases our knowledge on plant development and evolution, and may inspire future applications in crop species.

## 1. Introduction

Ferns constitute a diverse evolutionary lineage, sister to the seed plants, which is distributed throughout the world and plays an important role in ecosystem functioning. Although research on ferns is very limited compared to that on plants of interest to agroforestry, they have occasionally been considered to study basic developmental processes, such as photomorphogenesis [1], spore germination [2,3], cell polarity [4], cell wall biogenesis [5], etc. Specifically, fern gametophytes are ideal organisms for research on reproduction [6], which is facilitated by a simple in vitro culture system and their small size of a few millimeters [6]. Ferns can also provide some insights into adaptation to environmental changes; for instance, they survived periods with high CO_2_ levels [7]. In addition, ferns contain multiple secondary metabolites, including flavonoids, alkaloids, phenols, steroids, etc., and produce antibacterial, antidiabetic, anticancer, antioxidant, and other bioactive compounds used in the herbal drinks of traditional medicine or the modern form of nanoparticles [8].

The gametophyte of *Dryopteris affinis* (Lowe) Fraser-Jenk. ssp. *affinis* (hereafter referred to as *DA*) is a preferred model organism for investigating apomixis. This mode of asexual reproduction bypasses meiosis and fertilization to form embryos, and the introduction of apomixis into sexual crops is considered the ‘holy grail of agriculture’ by scientists around the world [9,10,11]. Apomixis is more frequent in ferns than in other plant groups [12]. In *DA*, spores give rise to gametophytes that form antheridia but not archegonia; thus, sexual reproduction is not feasible and the eggs in the archegonia form embryos without fertilization, the progeny being genetically identical to the mother plant. *DA* was formed by homoploid hybrid speciation between the sexual diploid *D. oreades* (hereafter referred to as *DO*) and an unknown ‘pure’ *D. affinis* ancestor [13]. Most spores of *DA* have the same (diploid) constitution as the sporophyte—and have thus not undergone meiosis—and germinate at similar rates to those of related sexual species [14].

The molecular basis of growth and development of *DA*—and ferns in general—has been elusive until recently because they exhibit higher chromosome numbers and larger genomes than mosses and seed plants, making it more difficult to obtain genomic data. Thanks to the advent of high-throughput technologies, it has been possible to perform transcriptomics by next-generation sequencing (NGS) and shotgun proteomics by tandem mass spectrometry over recent years. However, although there are now almost 100 studies on fern transcriptomes, the proteome has been studied in only a few species: *Azolla microphylla* [15]; *Ceratopteris richardii* [16], *Cyathea delgadii* [17], *Diplazium maximum* [18], *Lygodium japonicum* [19], *Mohria caffrorum* [20], *Pteris vittata* [21], *Pteridium aquilinum* [22,23], *Struthiopteris spicant* [24], and also in the apogamous *DA* and its sexual relative *DO* [25,26,27]. In this regard, we previously compared the protein profiles of gametophytes of *DA* and *DO* from a total of 879 quantifiable proteins. The goal of the present study was to extend the analyses conducted on the gametophytes of these species and explore potential protein interactions. Specifically, 417 proteins that were equally present in the gametophytes of both species and the protein–protein interactions within them were analyzed using the STRING platform (version 11.5).

## 2. Results

### 2.1. Identification of Proteins by LC-MS/MS

Our proteomic analyses involved a total of 417 quantifiable proteins (Appendix A) that were equally present in both apogamous (*DA*) and sexual (*DO*) gametophytes, exhibiting a total of three assigned spectra and rejecting those with only a few counts. Specifically, a subset of 145 proteins assigned to the functional groups ‘growth’, ‘reproduction’, ‘phytohormone signaling and biosynthesis’, ‘gene expression’, and ‘biotic and abiotic stress’, which are involved in growth, reproduction, phytohormones signaling, and the regulation of gene expression are reported here. The resulting proteome dataset might contribute to insights into the molecular mechanisms governing gametophyte development, a largely overlooked stage of the plant life cycle. Furthermore, the presented dataset increases the general knowledge about this underestimated group of plants.

### 2.2. Protein Analyses in Fern Gametophytes

To obtain better insights into the function of both collections of proteins, Gene Ontology (GO) classification analyses provided by the STRING platform were performed. The set of 417 proteins were checked according to the standard categories ‘biological function’, ‘molecular function’, and ‘cellular component’. In the category ‘biological processes’ (Figure 1a), the largest number of proteins shared by both species have a function in response to stress (about 25%). In addition, there were abundant proteins involved in the primary metabolism of amino acids, peptides, proteins (including folding, transport, and proteolysis), and nucleic acids. Several proteins involved in nitrogen and sulfur metabolism, ribosome biogenesis, gene expression, and translation were also included. We also found a large group of proteins that are involved in the generation of precursor metabolites, energy, and photosynthesis, as well as others involved in plant developmental processes. A smaller number of proteins is associated with translation, transport, toxin catabolism, and chaperone-mediated protein folding.

Under the category ‘molecular function’ (Figure 1b), a very high number of proteins are classified to have ‘binding activity’, mainly to organic cyclic compounds and ions, or to have ‘catalytic activity’. Proteins with oxidoreductase activity are very relevant, as are those related to ‘structural molecule activity’. Noteworthy also are the small number of chaperones detected, of proteins that make up the translation elongation factor, as well as of those with antioxidant activity which, despite being quite important for the detoxification of reactive oxygen species in the cell, represented less than 10% of the total proteins identified and quantified.

Under the category ‘cellular component’ (Figure 1c), most of the proteins were predicted to localize to the cytoplasm and intracellular organelles, especially the chloroplasts. In both species, similar amounts of proteins were found to be associated with the cell wall, plasmodesmata, and thylakoid and plasma membranes, whereas proteins present in the membranes of vesicles of the Golgi apparatus were scarce compared to other subcellular compartments. Likewise, we found quite a few proteins associated with the peroxisome, an organelle that performs very specific functions in plant development and reproduction.

Using the STRING database, an analysis of proteins shared by both species found 417 knots, 3471 interactions, and a probability of enrichment of protein–protein interactions of less than 10^−16^. These data indicate that proteins have more interactions with each other than might be expected for a random set of proteins of similar size, which means that the proteins are at least partially connected within a group. Here, we focused on a collection of 145 proteins associated with the functional groups for growth, reproduction, phytohormone signaling and biosynthesis, gene expression, response to biotic stress, and response to abiotic stress. The interactome is shown in Figure 2, and only a few proteins lacked any kind of connection. The sources of functional STRING interactions of the *DA* and *DO* proteins are detailed in Appendix A, and refer to protein–protein associations obtained from co-expression, co-occurrence, text mining, homology, databases, and experimental datasets. The interactions among the proteins are depicted in four groups: those of growth and reproduction, phytohormones, gene expression, and abiotic stress (Figure 3). In this regard, the majority of the proteins associated with growth and reproduction refers to aspects of anatomical structure and systems development. In the case of phytohormones, they are linked to the positive regulation of the abscisic acid (ABA)-activated signaling pathway, endocytosis, and the regulation of biological quality. Concerning gene expression, the proteins found are linked to chromatin organization and translation. Finally, abiotic stress is overrepresented by processes such as the biosynthesis of secondary metabolites, protein folding, glutathione metabolism, superoxide dismutase activity, or by structural components possessing a DnaJ domain.

Paying attention to the major source of STRING interactions in each functional group of proteins, some additional information can be extracted (Figure 4). For instance, the major source for interactions among proteins associated with growth and reproduction is text-mining data, while for phytohormones it is text mining and databases. Concerning gene expression, the major sources are co-expression data (Figure 5), which represents 50% of the score values, and experimental datasets.

The platform also provides a qualitative estimation of the interaction strength that these proteins have in the model species *A. thaliana* or in other organisms. For example, one can observe a strong interaction between two chloroplast proteins: the homologs of translation factor GUF1 (AT5G08650) and the protein elongation factor EF-G (SCO1) (Figure 6). Finally, in the case of proteins involved in abiotic stress, the interaction score value comes from text mining, homology, and phylogenetic co-occurrence data (Figure 6).

In fact, homology and co-occurrence are reported as the highest sources for the protein interaction scores in all groups of proteins selected for further analysis. Concerning phylogenetic co-occurrence, the taxa include bacteria, fungi, eudicotyledonous plants, the lycopod *S. moellendorffii*, the moss *Phycomitrium patens*, and several algae. As was mentioned above, the proteins of our two fern species were analyzed by STRING using the corresponding *A. thaliana* identifiers. Data in Figure 6 reveal that there are some proteins absent in bacteria: MITOCHONDRIAL PHOSPHATE CARRIER PROTEIN 3 (PHT3.1), LATE EMBRYOGENESIS ABUNDANT 7 (LEA7), EUKARYOTIC TRANSLATION ELONGATION FACTOR 5A-3 (ELF5A-3), RIBOSOMAL PROTEIN L10B (RPL10B), and AT1G05520), while fungi miss TRANSALDOLASE 2 (TRA2), LEA7, STARCH EXCESS1 (SEX1); and FIBRILLIN (FIB), and LEA7 is neither present in *S. moellendorffii* nor in *P. patens*. We did not manage to identify GLUTATHIONE S-TRANSFERASE 19 (GSTU19) in moss, and there are some additional gaps for the proteins TRA2, AT2G18340, LEA7, CATALASE (CAT), CBS DOMAIN CONTAINING PROTEIN 3 (CBSX3), COPPER/ZINC SUPEROXIDE DISMUTASE 1 (CSD1), THIAMINE THIAZOLE SYNTHASE 1 (THI1), GSTU19, and AT1G05520.

## 3. Discussion

Out of 879 quantifiable proteins [27], we detected a large number (417) that are shared by the gametophytes of apomictic *DA* and its sexual parent *DO*. These results, together with a previous study using the same two species [27], enlarge the proteomic profile of both ferns. However, unlike for *DA* [25], transcriptome data for *DO* are lacking, such that most certainly fewer proteins could be identified in *DO* than in *DA* due to sequence divergence. The classification of proteins according to biological processes shows that those associated with responses to abiotic and biotic stresses are predominant in both species (about 25% of all identified proteins). Specifically, we paid attention to a subset of 145 proteins falling into the six functional categories, i.e., growth, reproduction, phytohormone signaling and biosynthesis, gene expression, biotic stress, and abiotic stress (Table 1). We discuss their putative roles in plant development, followed by an analysis of predicted protein interactions.

### 3.1. Growth

The protein SHEPHERD (SHD) is located in the endoplasmic reticulum and belongs to the HtpG chaperone proteins, which are part of the HSP90 heat shock protein (HSP) family. This protein is involved in regulating the size and organization of the meristem. Single and double mutant analyses suggest that SHD may be required for the correct folding and/or complex formation of CLAVATA (CLV) proteins, as mutations of this family in *A. thaliana* exhibit expanded shoot meristems, disorganized root meristems, and defective pollen tube elongation [28]. Three proteins are related to the regulation of cell division: CELL DIVISION CONTROL PROTEIN 48 HOMOLOG A (CDC48A) interacts with certain SNAREs as part of specialized membrane fusion events [29]; ACTIN7, an essential component of the cytoskeleton, is a vegetative actin with a role in hormone-induced cell proliferation and callus formation [30]; and the P(1B)-Type ATPase1 (PAA1), a copper-transporting ATPase, has a role in cell proliferation, in this case of glucose signaling-regulated root meristem development [31]. The list also includes the dose-dependent CELL DIVISION PROTEIN FtsZ HOMOLOG 1 (FTSZ1) protein, a component of the plastidial division machinery, which is also involved in blue light-induced chloroplast movement [32].

The gametophytes of both species are characterized by the presence of abundant trichomes, which might be related to the identification of ENOLASE1 (ENO1), an enolase involved in glycolysis, mutants of which produce abnormal trichomes in *A. thaliana* [33]. The PSBQ-2 protein, a subunit of the oxygen-evolving complex of photosystem II, could help to cope with growth in low-light conditions and/or to assemble and stabilize photosystem II [34]. The GLUTATHIONE S-TRANSFERASE TAU 20 (GSTU20) protein is also involved in—among others processes—regulating the influence of far-red light on development and probably gravitropic signal transduction [35,36,37]. Finally, we identified two chaperones: CHAPERONIN CONTAINING TCP1 SUBUNIT 8 (CCT8), contributing to the maintenance of stem cells [38], and CHLOROPLAST STEM-LOOP BINDING PROTEIN 41 kDa (CSP41B), involved in the regulation of the circadian rhythm [39]. Finally, CYCLOPHILIN-40 (CYP40), a peptidyl-prolyl *cis-trans* isomerase, promotes the juvenile phase of vegetative development and—to a lesser extent—the positioning of floral buds, floral morphogenesis, and the expression of heat shock proteins [40].

### 3.2. Reproduction

Although ferns do not form flowers or seeds, we found proteins related to the formation of these reproductive organs in flowering plants. This finding suggests that the genes encoding those proteins arose in evolution before the divergence of the fern and seed plant lineages. This stresses the importance of carrying out genomic and proteomic studies from the perspective of evolutionary developmental biology. Below, we highlight some of the most striking proteins that are associated with gametophyte development, embryogenesis, or flowering.

Among the proteins identified in our fern gametophytes, there are some that impair pollen development in seed plants if defective, either by influencing metabolic or structural processes. Those affecting metabolic processes include the cytosolic isoform of PHOSPHOGLUCOMUTASE 2 (PGM2); EMBRYO SAC DEVELOPMENT ARREST9 (EDA9), encoding a D-3-phosphoglycerate dehydrogenase; the catalytic SUBUNIT A OF VACUOLAR ATP SYNTHASE (VHA-A); PURINE BIOSYNTHESIS4 (PUR4), a probable phosphoribosyl formylglycinamidine synthase; and the REVERSIBLY GLYCOSYLATED POLYPEPTIDE1 (RGP1). Proteins influencing structural processes include those involved in actin filament bundling in pollen tubes, such as PROFILIN5 (PRF5) and VILLIN 2 (VLN2), a Ca^2+^-regulated actin-binding protein. In addition, two tRNA ligases required for female gametophyte development were found: OVULE ABORTION9 (OVA9) and OVA4, involved in the aminoacylation of tRNAs [41].

Embryogenesis is central to reproduction of any organism. In ferns, gametophytes form the egg cells that, upon fertilization (sexuals) or independent of it (apomicts), produce the sporophytic embryo. Homologs of several proteins we identified are involved in embryogenesis in seed plants, including CHLORATE RESISTANT88 (CR88), encoding the heat shock protein HSP90.5, a chaperone essential for the biogenesis and maintenance of chloroplasts [42]; the CYCLOPHYLLIN 19 (CYP19-4) protein, which interacts in vitro with GNOM, a GDP/GTP exchange factor that plays a central role in embryonic pattern formation [43]; and the PREPHENATE AMINOTRANSFERASE (PAT), a maternally required aspartate aminotransferase necessary for the development of the embryo [44]. In addition, we identified EMBRYO DEFECTIVE2761 (EMB2761), a threonyl-tRNA synthetase found in mitochondria and chloroplasts, that—when affected by a mutation—causes embryonic arrest at the globular stage [41].

As for proteins related to flowering, the fern gametophytes expressed FLOWERING LOCUS D (FLD), a histone demethylase that promotes flowering regardless of photoperiod and vernalization by repressing *FLOWERING LOCUS C* (*FLC*) that, in turn, encodes a floral repressor blocking the transition from vegetative to reproductive development [45]. Likewise, we identified EMB14, the pre-mRNA processing-splicing factor 8A, which plays an important role in embryonic development and regulates the splicing efficiency of the COOLAIR non-coding RNA at the *FLC* locus [46].

### 3.3. Phytohormone Signaling and Biosynthesis

In the gametophytes of our two fern species, we identified proteins related to the major families of phytohormones, for instance auxins, expanding our previous results in *DA* [25,26]. The RNA-binding protein AT1G76940, which modulates the effect of auxin on the transcriptome [47], is worth mentioning. We also found SHORT-CHAIN DEHYDROGENASE/REDUCTASE A (SDRA), involved in the β-peroxisomal oxidation of IBA to form IAA. IAA is biologically active and could be involved in the peroxisomal activation of 2,4-D, a precursor of the auxins that inhibit root growth [48]. In line with this, another interesting protein is the chaperone BOBBER1 (BOB1), belonging to the HSP20-like superfamily of non-canonical small heat shock proteins, which is necessary for the establishment of auxin gradients, the patterning of the apical domain of the embryo, and other developmental processes [49]. We also detected three other common proteins related to auxin: CLATHRIN HEAVY CHAIN 2 (CHC2) is required for auxin polar transport through regulating the distribution of transporters of the PIN-FORMED (PIN) family [50]; ADP-RIBOSYLATION FACTOR A1F (ARFA1F), belonging to the family of ARF GTPases, is involved in cell division, cell expansion, and cellulose production [51]; and CHALCONE SYNTHETASE (CHS), a key enzyme involved in the biosynthesis of flavonoids, including the purple anthocyanins that accumulate in leaves and stems, participates in the regulation of auxin transport, and modulates of root gravitropism [52].

We also identified some proteins related to the phytohormones ABA and cytokinin, such as the UDP-GLYCOSYLTRANSFERASE 85A1 (UGT85A1), involved in the O-glycosylation of *trans*-zeatin and dihydrozeatin, and active in vitro on *cis*-zeatin. In addition, ABA DEFICIENT1 (ABA1), a zeaxanthin epoxidase, plays an important role in the xanthophyll cycle and ABA biosynthesis by converting zeaxanthin into antheroxanthin and subsequently into violaxanthin. This protein is necessary for many important processes in plants, including resistance to osmotic stress and drought, ABA-dependent closure of stomata, seed development and dormancy, modulation of defense gene expression, as well as resistance to disease and non-photochemical quenching [53,54]. Moreover, we identified the non-specific LIPID-TRANSFER PROTEIN 4 (LTP4), also regulated by ABA. This protein binds to fatty acids and acyl-CoA esters and can transfer several different phospholipids. Its biological function can be linked to stress by salt or water deprivation. In ferns, lipid composition also plays an important role in the integrity and fluidity of membranes and in desiccation tolerance [55].

As for gibberellin-related proteins, TOPLESS-RELATED PROTEIN 4 (TRP4) is a transcriptional co-suppressor involved in the regulation of homeostasis and signaling of these phytohormones. The founding member of the TOPLESS family regulates the polarity of the embryo, preventing the formation of a root meristem in the apical domain [56], and acts redundantly with other TRP proteins.

We found two proteins outside the main families that are relevant for their relationship with important compounds that are little studied in seed plants and even less so in ferns. One is O-METHYLTRANSFERASE 1 (OMT1), involved in methylating flavonols and the biosynthesis of melatonin. The other is GLUTAMATE DECARBOXYLASE 1 (GAD1), which catalyzes γ-aminobutyric acid (GABA) production. The non-protein amino acid GABA has multiple functions under both stress and non-stress conditions and in reproduction [57,58].

### 3.4. Gene Expression

Undoubtedly, proteins regulating gene expression play a central role in biological processes by participating, in one way or another, in the conversion of genetic into metabolic information. We found many proteins common to *DA* and *DO* that are involved in methylation, histone deacetylation, gene silencing, unwinding of nucleic acids, transcription activation or repression, etc. Among the proteins related to gene silencing, we identified HYPONASTIC LEAVES1 (HYL1), which forms a complex with SERRATE and DICER-LIKE1 (DCL1), to promote the precise processing of pre-mRNA by DCL1, and with ARGONAUTE1 (AGO1), which is involved in post-transcriptional gene silencing [59,60,61]. Furthermore, we found HOMOLOGY-DEPENDENT GENE SILENCING1 (HOG1), a S-adenosyl-L-homocysteine hydrolase required for—among other functions—transcriptional gene silencing dependent on DNA methylation.

We also identified some histones, the core components of nucleosomes. DNA is wrapped around nucleosomes, which can compact chromatin, limiting the accessibility of DNA to the cellular machinery that requires DNA as a template. Therefore, histones play a central role in transcriptional regulation, DNA repair, DNA replication, and chromosomal stability. Accessibility to DNA is regulated through a complex set of post-translational histone modifications, also called the ‘histone code’, and nucleosome remodeling, whereby nucleosomes are moved along, unwrapped, or ejected from the DNA. We found HISTONE DEACETYLASE 3 (HDT3), a HD2-type deacetylase, which likely mediates the deacetylation of lysine residues at the amino terminal end of the core of histones (H2A, H2B, H3, H4). Histone deacetylation acts as a label for epigenetic repression and plays an important role in transcriptional regulation, cell cycle progression, and various developmental events. Likewise, this process was suggested to be involved in the modulation of ABA signaling and the regulation of stress-responsive genes [62]. HISTONE H2A PROTEIN 9 (HTA9) is a histone H2A variant that is synthesized throughout the cell cycle in contrast to the canonical S phase-regulated H2A. Another protein worth mentioning is RESISTANCE TO PSEUDOMONAS SYRINGAE PV MACULICOLA INTERACTOR1 (RIN1), a central component of the INO80 chromatin remodeling complex that is involved in transcriptional regulation, DNA replication, and probably DNA repair.

In addition, we found the transcription factor PURIN-RICH ALPHA1 (PURA1), which binds specifically to repeated purine-rich sequences found in *cis*-regulatory elements called teloboxes. Several helicases, involved in the unwinding of DNA and/or RNA, have also been identified: RNA HELICASE 15 (RH15), a DEAD-box helicase involved in processing of pre-mRNA and nuclear export, and RH37. Finally, we mention UBIQUITIN-CONJUGATING ENZYME 35 (UBC35) that is associated with DNA repair upon damage by UV radiation [63].

### 3.5. Biotic Stress

Plants must continuously cope with attacks by other macro- and microorganisms. We identified some proteins with a possible role in preventing damage and counteracting, in some way, their vulnerable condition due to sessility. Among others, we detected the protein SUPPRESSOR OF G2 ALLELE OF *skp1* HOMOLOG B (SGT1B), involved in the innate immunity of plants [64] and essential for the resistance conferred by multiple resistance (*R*) genes that recognize different isolates of oomycete pathogens, such as the downy mildew *Hyaloperonospora parasitica* [65]. Another protein of interest is 3-DEOXY-D-ARABINO-HEPTULOSONATE 7-PHOSPHATE SYNTHASE 1 (DHS1), which is induced by wounding and infections with the pathogenic bacterium *Pseudomonas syringae* [66]. On the other hand, the SUPPRESSOR OF SA INSENSITIVE2 (SSI2) is a desaturase that can intervene in defense signaling mediated by salicylic acid (SA) and jasmonic acid (JA). The *ssi2* mutant has an increased 18:0 and reduced 18:1 fatty acid composition and shows hyper-resistance to the green peach aphid and antibiotic activity in the exudate of the petiole [67]. Membrane lipid biosynthesis is a crucial process, and the ACYL CARRIER PROTEIN 4 (ACP4) plays an essential role in fatty acid biosynthesis in chloroplasts, the formation of cuticular waxes and cutin polymers in leaves, as well as in the establishment of systemic acquired resistance [68]. Moreover, PHOSPHOLIPASE D ALPHA1 (PLDALPHA1) hydrolyzes the terminal phosphodiester bond of glycerol-phospholipids to generate phosphatidic acids, and plays an important role in several cellular processes, including the action of phytohormones and the stress response to cellular acidification. PLDALPHA1 is involved in the induction of JA upon wounding [69] and probably in freezing tolerance by modulating cold-induced gene expression and osmolite accumulation [70]. The protein also mediates the effects of ABA on stomata, on seed aging and deterioration, as well as on microtubule stabilization and salt tolerance [71]. In line with the abovementioned, we found PEROXISOMAL 3-KETOACYL-COA THIOLASE 3 (PKT3), which influences the induction of JA biosynthesis after injury and during senescence [72]. We have already commented on some proteins related to the synthesis of GABA, but we also identified IMPAIRED IN BABA-INDUCED DISEASE IMMUNITY1 (IBI1), an aspartate-tRNA ligase involved in the perception of β-aminobutyric acid (BABA) that is required for the priming effect of BABA in disease resistance [73]. Besides, we identified OPEN STOMATA1 (OST1), a SNF1-related serine/threonine protein kinase that is activated by the ABA signaling pathway and regulates numerous ABA responses, including stomatal closure in response to drought, pathogens, or a decrease in humidity [74]. On the other hand, NONHOST RESISTANCE TO P. S. PHASEOLICOLA1 (GLPK) performs a limiting step in glycerol metabolism and is necessary for both general and specific resistance to bacteria and fungi [75]. Finally, we want to mention the identification of the metallopeptidase ORGANELLAR OLIGOPEPTIDASE (OOP), which was reported to bind SA [76].

### 3.6. Abiotic Stress

Fern gametophytes harbor a huge biochemical arsenal to cope with abiotic stimuli in order to preserve their reproductive function. Several factors are likely to modify the redox state of this vulnerable generation of the life cycle. In reviewing our proteome profiles, we found numerous proteins involved in “oxidative stress”, such as the cytosolic COPPER/ZINC SUPEROXIDE DISMUTASE 1 (CSD1), the expression of which is affected by mRNA cleavage directed by a microRNA (miR398); the MANGANESE SUPEROXIDE DISMUTASE 1 (MSD1); and two thiol-specific peroxiredoxins: PERIREDOXIN Q (PRXQ) and chloroplastic 2-CYSTEINE PEROXIREDOXIN BAS1-like (2CPB). Other proteins grouped here are DEHYDROASCORBATE REDUCTASE 3 (DHAR3), a key component of the glutathione S-transferase ascorbate recycling system; GLUTATHIONE S-TRANSFERASE PHI 10 (GSTF10) and GSTU19, which detoxify certain herbicides; and a MONODEHYDRO ASCORBATE REDUCTASE (MDAR), which converts monodehydroascorbate to ascorbate, oxidizing nicotinamide adenine dinucleotide hydrogen (NADH) to NAD^+^ in the process.

Identified proteins involved in the response to “light intensity” include SERINE HYDROXYMETHYLTRANSFERASE 1 (SHM1), which acts in damage caused by sodicity and the hypersensitive defense response [77], and a GroES-like zinc-binding alcohol dehydrogenase that functions as an oxidoreductase encoded by *AT5G61510*. In addition, we found FIBRILLIN (FIB), which interacts with ABA-INSENSITIVE2 (ABI2) [78] and is involved in ABA-mediated photoprotection. Moreover, we found the RIBOSOMAL PROTEIN L10B (RPL10B), a structural constituent of the ribosome that might participate in the response to UV-B [79].

Regarding stress caused by “salts and heavy metals”, we identified O-ACETYLSERINE (THIOL) LYASE ISOFORM A1 (OASA1), which is induced by high salt and heavy metal concentrations in an ABA-mediated fashion [80]; the transporters PHOSPHATE TRANSPORTER 3;1 (PHT3;1), and the protein cystathionine β-synthase CBS DOMAIN CONTAINING PROTEIN 3 (CBSX3), a cobalt ion binding protein affecting plant growth and reproduction [81].

Associated with “drought and temperature”, we identified the protein LATE EMBRYOGENESIS ABUNDANT7 (LEA7), which might protect plants by avoiding the aggregation of soluble proteins in leaves [82]. Besides, there was LOW EXPRESSION OF OSMOTICALLY RESPONSIVE GENES4 (LOS4), a putative DEAD-box RNA helicase thought to act in temperature sensing [83]. In line with this, the co-chaperone FK506 BINDING PROTEIN 62 (FKBP62) positively modulates thermotolerance by interacting with HSP90.1, as well as increasing HEAT SHOCK TRANSCRIPTION FACTOR A2 (HSFA2)-mediated accumulation of chaperones of the small-HSP family. CHAPERONIN 10 (CPN10) is also upregulated in response to a heat shock treatment. In gametophytes of both *DA* and *DO*, several proteins belonging to the HSP20, HSP70, and HSP90 families were identified: HSP 17.6II, HSP 21, MTHSC70-2, HSP 70 KDA, HSP70-15, HSP 81.4, and HSP91, which protect from heat stress damage through their ability to recognize non-native conformations of other proteins. In addition, we detected PEPTIDE METHIONINE SULFOXIDE REDUCTASE (PMSR), which prevents the methionine sulfoxidation of HSP21 and thus its inactivation. The list continues with other chaperones, such as the CASEIN LYTIC PROTEINASE B3 (CLPB3), a HSP101 homolog that confers thermotolerance to chloroplasts during heat stress [84], and DNA J PROTEIN A6 (DJA6), which may function together with the HEAT SHOCK COGNATE PROTEIN 70 (HSC70) chaperone to assist protein folding and prevent protein aggregation during heat stress in the chloroplast [85]. Additionally, two proteins related to cold stress conditions were found: the 31-KDA RNA BINDING PROTEIN (RBP31), which stabilizes transcripts of numerous mRNAs and modulates telomere replication [86], and the alpha-glucan water dikinase STARCH EXCESS1 (SEX1) [87].

Involved in “response to cadmium ions”, gametophytes of both species express some oxidoreductases: a probable N-ACETYL-GAMMA-GLUTAMYL-PHOSPHATE REDUCTASE (AT2G19940), a putative NADPH-DEPENDENT MANNOSE 6-PHOSPHATE REDUCTASE (AT2G21250), and two pyruvate kinases: AT3G52990 and AT5G63680. 

We also report proteins related to “vitamin metabolism”. The pyridoxal 5’-phosphate synthase subunit PYRIDOXINE BIOSYNTHESIS 1.3 (PDX1.3) plays several roles in vitamin B6 biosynthesis, development, stress tolerance, and resistance to singlet oxygen-generating photosensitizers in *A. thaliana* [88]. In addition, the thiazole biosynthetic enzyme THIAMINE1 (THI1), defined as a suicide enzyme because it undergoes only a single turnover, is involved in the biosynthesis of the thiamine precursor thiazole, and the adaptation to various stress conditions [89]. Other noteworthy proteins are the chloroplast protein TRANSKETOLASE 1 (TKL1), linked to the pentose-phosphate shunt, which may also act as a stress sensor [90]; the SULPHURTRANSFERASE 1 (STR1), involved in embryo and seed development [91]; and the EUKARYOTIC ELONGATION FACTOR 5A-3 (ELF5A-3), which acts as a bimodular protein capable of binding to both RNA and proteins and supports growth in response to sublethal osmotic and nutrient stresses [92].

### 3.7. Protein–Protein Interactions

The network of proteins clustered inside the group “growth and reproduction”, has a PPI enrichment *p*-value of 5.55 × 10^−16^, meaning that the proteins are at least partially biologically connected as a group. We focus on those proteins associated with anatomical structure, organelle organization, and systems development, i.e., any process that modulates a qualitative or quantitative biological trait such as an assessable attribute of an organism or part of an organism (size, mass, shape, color, etc.). The proteins RIBULOSE-5-PHOSPHATE-3-EPIMERASE (RPE) and EDA9 exhibit maximum interactions with 12 and 11 nodes, respectively, followed by ENO1, GLYCERALDEHIDE-3-PHOSPHATE-DESHIDROGENASE-1 (GAPCP-1), PHOSPHOGLUCOMUTASE (PGMP), SERINEHYDROXYMETHYLTRANSFERASE 4 (SHM4), GLYCERALDEHYDE-3-PHOSPHATE DESHYDROGENASE (GAPCP-2), PGM2, SHD, OVA4, and PUR4, all of them having more than six nodes. The strength of each type of interaction between proteins is expressed by a scale provided by the STRING platform, ranging from 0 to 1, with 1 meaning maximal and 0 minimal interaction, respectively. According to this scale, the highest interaction score values are those involving the connection between the protein ENO1 and the proteins GAPCP-1, GAPCP-2, PGM2, and PGMP. A very high score is found between PGMP and PGM2, from accurate databases. The values come from co-expression data. Enolases are enzymes that catalyze the reversible conversion of D-2-phosphoglycerate into phosphoenol pyruvate in the glycolytic pathway, which contributes to satisfy the energy demand of the organism. In fern gametophytes, similar to pollen formation and germination in flowering plants, metabolic pathways involving mitochondrial respiration and fermentation are expected to play a very important role [93]. Three subnetworks more are observed in the growth and reproduction set of proteins. First, the interactions among ACT7 with VLN2 and PRF5, important for actin filament bundling and the organization of the actin cytoskeleton, and also among OVA4, OVA 9, and EMB276, which are t-RNA ligases involved in translation and associated with reproduction [41,94]. Second, the protein SHM4, a probable phosphoribosyl-formylglycinamidine-synthase that is essential to male gametophyte development in *A. thaliana*, and PUR4, a glycine-hydroxymethyl-transferase, have homologs that are neighbors, either co-expressed or mentioned together in other organisms. Third, there are the connections among AT3G03960, CPN60A, SHD, and SQN. Specifically, the interaction between SHD and SQN has already been reported in text-mining datasets [95].

Proteins related to “phytohormones” show two strong interactions: on the one hand, it is the couple ARFA1F-AT3G0830, the first an ARF GTPase essential for vesicle coating and uncoating, and the second mediating endocytosis, which is involved in the correct distribution of PIN auxin transporters. High score values were obtained from co-expression, experimental data, association in curated databases, and text-mining evidence. The other important interaction is between IBRI, a dehydrogenase/reductase involved in the peroxisomal β-oxidation of the auxin IBA to form the biologically active IAA, and PKT3, a 3-ketoacyl-CoA thiolase involved—among other functions—in long chain fatty-acid β-oxidation prior to gluconeogenesis during germination and subsequent seedling growth. Interaction data are provided from co-expression, experimental data, association in curated databases, and text-mining information [96].

The interactions between proteins associated with “gene expression” show 21 nodes, 26 edges, and a PPI enrichment *p* value of 3.19 × 10^−7^. In this list of proteins, we can identify two subnetworks. One of them contains some elongation factors, turning around AT1G09640, such as EIF4A1, AT2G40290, SCO1, and other proteins required for DNA methylation-dependent gene silencing, such as HOG1. The second subnetwork connects proteins related to nucleosome components, such as the histones AT1G09200, T11P11.4, HTB1, RIN1, and the helicase UAP56a. The highest score values come from co-expression, for instance for the following protein pairs: AT1G57660–AT3G60245, HOG1–UAP56a; RIN1–UAP56a; AT1G09640–AT1G57660; AT2G40290–EIF4A1; AT5G08650–SCO1; and HTA9 – UAP56a. All of them have score values above 0.839. These results are expected by the prevalence of biological functions linked to chromatin structure and regulation.

The proteins involved in “response to abiotic factors” were predominant in the studied gametophytes, representing up to 25% of the total proteins. In the corresponding network, there are 59 nodes and 84 edges, and a PPI enrichment *p* value of <1.0 × 10^−16^. The proteins are mostly related to protein folding, DnaJ domain and HSP90 proteins, glutathione metabolism, superoxide dismutase activity, redox active center, and the biosynthesis of secondary metabolites. The proteins CAT, GPX7, MTHSC70-2, TRA2, CSD1, PRXQ, AT5G06290, CPN10, CR88, HSP70-15, and HSP81.4 have between 5 and 11 nodes. According to STRING data, there are co-expressed proteins such as AOR–PRXQ; AT3G60750–TRA2; ELF5A-3–RPL10B; AT3G29320–SEX1; Hsp70-15–MTHSC70-2; AT5G06290–PRXQ; MTHSC70-2–ROF1; At1G53540–HSP17.6II; and PRXQ–SHM1. Similarly, the high scores for protein interactions were obtained from information extracted from curated databases. This is also the case for the following pairs: AOR–PRXQ; AT3G60750–TRA2; CAT–GOX1; ALDH5F1–AT5G14590; ALDH5F1–cICDH; RSR4–TRA2; and AT3G60750–RSR4. In addition, strong interactions were obtained from text mining between: CAT–CSD1; AT5G06290–TRX-M4; AT5G06290–PRXQ; GSTF10–GSTU19; GSTF9–GSTU19; AT1G53540–HSP17.6II; and AT5g06290–GPX7, and, finally, from existing gene fusion events, between the aldolase AT3G60750 and the transketolase TRA2 in *Proteobateria* and *Actinobacteria*. Both proteins contribute to establish a link between the glycolytic and pentose phosphate pathways. Indeed, transaldolases derived from *Actinobacteria* are present in land plants, suggesting they supported vascular development and the adaptation to life on land [97]. However, this gene transfer has not been studied in ferns, and it should be given more consideration to gain insights from evo-devo studies in plants. Likewise, co-occurrence data provided by the STRING platform should raise interest in this type of analysis to shed light onto the molecular basis of plant developmental processes. These data show that most proteins involved in the response to abiotic stresses correlate across the phylogenetic tree, as if they were a transversal legacy of genes having been under common selective pressure during evolution and may, therefore, be functionally associated. Further research should be done to clarify the considered interactions and inferences.

## 4. Materials and Methods

### 4.1. Plant Material and Growth Conditions

Spores were obtained from two populations in northern Spain: *DA* in Turón valley (Asturias, 43°12′10″ N−5°43′43″ W) and *DO* in Neila lagoons (Burgos province, 42°02′48″ N−3°03′44″ W). For each population, ten sporophytes were randomly selected and a frond with sporangia was collected from each individual. Spores were released from sporangia by drying the fronds on sheets of smooth paper for 1 week in the laboratory. Spores from the ten individuals of each population were then pooled, soaked in water for 2 h, and dipped for 10 min in a solution of NaClO (0.5%) and the tensioactive agent Tween 20 (0.1%). Then, the remaining compounds were eliminated with sterile, distilled water. A change of solutions or water was performed by centrifugation at 1300× *g* for 3 min. Once sterilized, spores were cultured in 500-mL Erlenmeyer flasks containing 100 mL of liquid Murashige and Skoog (MS) medium [98]. Unless otherwise noted, media were supplemented with 2% sucrose (*w/v*), and the pH adjusted to 5.7 with 1.0 or 0.1 N NaOH. The cultures were kept at 25 °C under cool-white, fluorescent light (70 µmol m^−2^ s^−1^) with a 16 h photoperiod and placed on an orbital shaker (75 rpm).

Following spore germination, gametophytes showed filamentous (one-dimensional) growth. Then, they were sub-cultured into 200 mL flasks containing 25 mL of MS medium supplemented with 2% sucrose (*w/v*) and 0.7% agar. *DA* gametophytes become two-dimensional, reaching the spatulate and heart stages after 20 or 30 additional days, respectively. *DO* gametophytes grow slower and needed around six months to become heart-shaped and reach sexual maturity. Reproductively mature gametophytes were collected under a dissection microscope. Gametophytes were considered mature if they had archegonia (*DO*) or an apogamic center composed of smaller and darker isodiametric cells (*DA*) (Figure 7). The collected gametophytes were weighed before and after lyophilization for 48 h (Telstar-Cryodos) and stored in Eppendorf tubes in a freezer at −20 °C until used.

### 4.2. Protein Extraction

From the heart-shaped apogamous or sexual gametophytes (three samples each), 20 mg (dry weight) of gametophytes was ground using a Silamat S5 shaker (Ivoclar Vivadent, Schaan, Liechtenstein). Samples were solubilized with 800 μL of buffer A (0.5 M Tris-HCl (pH 8.0), 5 mM EDTA, 0.1 M HEPES-KOH, 4 mM DTT, 15 mM EGTA, 1 mM PMSF, 0.5% (*w/v*) PVP, and 1× protease inhibitor cocktail (Roche, Rotkreuz, Switzerland)) were homogenized using a Potter homogenizer (Thermo Fisher Scientific, Bremen, Germany).

Proteins were extracted in two steps: first, the homogenate was subjected to centrifugation at 16,200× *g* for 10 min at 4 °C on a tabletop centrifuge and, second, the supernatant was subjected to ultracentrifugation at 117–124 kPa (∼100,000× *g*) for 45 min at 4 °C in an Airfuge (Beckman Coulter, Pasadena, CA, USA), yielding the soluble protein fraction in the supernatant. In parallel, the pellet obtained from the first ultracentrifugation was re-dissolved in 200 μL of buffer B (40 mM Tris-base, 40 mM DTT, 4% (*w/v*) SDS, 1 × protease inhibitor cocktail (Roche, Rotkreuz, Switzerland)) to extract membrane proteins using the ultracentrifuge as described above, in the supernatant. Protein concentrations were determined using a Qubit Fluorometer (Invitrogen, Carlsbad, CA, USA). The 1D gel electrophoresis was performed, and 1 mg protein per each soluble and membrane fraction was treated with sample loading buffer and 2 M DTT, heated at 99 °C for 5 min, followed by a short cooling period on ice, and then loaded separately onto a 0.75 mm tick, 12% SDS-PAGE mini-gel. Electrophoresis conditions were 150 V and 250 mA for 1 h in 1× running buffer.

### 4.3. Protein Separation and In-Gel Digestion

Each gel lane was cut into six 0.4 cm wide sections resulting in 48 slices, then fragmented into smaller pieces and subjected to 10 mM DTT (in 25 mM AmBic, pH8) for 45 min at 56 °C and 50 mM iodoacetamide for 1 h at room temperature in the dark, prior to trypsin digestion at 37 °C overnight [99]. Subsequently, they were washed twice with 100 μL of 100 mM NH_4_HCO_3_/50% acetonitrile and washed once with 50 μL acetonitrile. At this point, the supernatants were discarded. Peptides were digested with 20 μL trypsin (5 ng/μL in 10 mM Tris/2 mM CaCl_2_, pH 8.2) and 50 μL buffer (10 mM Tris/2 mM CaCl_2_, pH 8.2), and after microwave-heating for 30 min at 60 °C, the supernatant was removed, and gel pieces were extracted once with 150 μL 0.1% TFA/50% acetonitrile. All supernatants were put together, then dried and dissolved in 15 μL 0.1% formic acid/3% acetonitrile, and finally transferred to auto-sampler vials for liquid chromatography (LC)-tandem mass spectrometry (MS/MS) for which 5 μL was injected.

### 4.4. Mass Spectrometry and Peptide Identification (Orbitrap XL)

The samples were analyzed on a LTQ Orbitrap mass spectrometer (Thermo Fisher Scientific, Bremen, Germany) coupled to an Eksigent Nano HPLC system (Eksigent Technologies, Dublin, CA, USA). For this purpose, samples were dissolved in 3% acetonitrile/0.1% formic acid, the solvent composition of buffers being as noted: buffer A was 0.2% formic acid/1% acetonitrile, and buffer B 0.2% formic acid/99.8% acetonitrile. Peptides were loaded onto a self-made tip column (75 μm × 80 mm) packed with reverse phase C18 material (AQ, particle size 3 μm, 200 Å) (Bischoff GmbH, Leonberg, Germany), and eluted at a flow rate of 200 nL per min. after applying the ensuing LC gradient: 0 min.: 5% buffer B, 56 min.: 40% B, 60 min.: 47% B, 64 min.: 97% B, 71 min.: 97% B. Mass spectra were obtained in the m/z range 300–2000 in the Orbitrap mass analyzer at a resolution of 60,000 at m/z 400. MS/MS spectra were acquired in a data-dependent manner from the five most intense signals in the ion trap, using 28% normalized collision energy and an activation time of 30 ms. The precursor ion isolation width was fixed to m/z 3.0. Charge state screening was enabled, and singly charged precursor ions and ions with undefined charge states were omitted. Precursor masses already selected for MS/MS acquisition were left out from further selection for 120 s. MS/MS spectra were transformed to the Mascot generic format (.mgf) using MascotDistiller 2.3.2, and submitted to Mascot (Matrix Science, London, UK; version 2.4.01) for searching, trypsin being selected as the proteolytic enzyme. Mascot was arranged to search against the inhouse generated SSTDB (forward entries: 330,049) combined with the publicly available VPDB (forward entries: 1,031,407, downloaded from uniprot.org in March 2012), and a set of 260 known mass spectrometry contaminants in a target-decoy strategy (using reversed protein sequences). The concatenate dDB is available online (http://fgcz-r-021.uzh.ch/fasta/p1222_combo_NGS_n_Viridi_20160205.fasta (accessed on 9 November 2022)). Data were searched with a fragment ion mass tolerance of ±0.6 Da and a precursor mass tolerance of ±10 ppm, with a maximum of 2 missed cleavages being allowed. Carbamidomethylation of cysteine was specified as a fixed modification, and deamidation (N, Q), Gln->pyro-Glu (N-term Q), and oxidation (M) were specified in Mascot as variable modifications.

### 4.5. Protein Identification, Verification, and Bioinformatic Downstream Analyses

Mass spectrometry and peptide identification (Orbitrap XL) were performed according to Grossmann and colleagues [36]. Scaffold software (version Scaffold 4.2.1, Proteome Software Inc., Portland, OR, USA) was used to validate MS/MS-based peptide and protein identifications. Mascot results were analyzed together using the MudPIT option. Peptide identifications were accepted if they scored better than 95% probability as specified by the PeptideProphet algorithm [100] with delta mass correction, and protein identifications were accepted if the ProteinProphet [101] probability was above 95%. Proteins that contained the same peptides and could not be differentiated based on MS/MS alone were grouped to satisfy the principles of parsimony using the Scaffolds cluster analysis option. Only proteins that met the above criteria were considered as positively identified and used for further analyses. The number of random matches was evaluated by comparing the Mascot searches against a database containing reversed decoy entries and checking how many decoy entries (proteins or peptides) passed the applied quality filters. The peptide FDR and protein FDR were estimated at 2% and 1%, respectively, indicating the stringency of the analyses. A log_2_ ratio of the averaged spectral counts from both species was calculated, and proteins were considered to be differentially present if this ratio was above 0.99. In total, 417 proteins were equally present in gametophytes of *DA* and *DO*. To gain a functional understanding of the identified proteins, we blasted the whole protein sequences against *Selaginella moellendorffii* and *Arabidopsis thaliana* Uniprot sequences, and retrieved the best matching identifier from each of them, along with the corresponding e-value, accepting blast-hits with e-values below 1 × 10^−7^. These orthologue identifiers were then used in further downstream analyses.

### 4.6. Protein Analysis Using the STRING Platform

The identifiers of the proteins from the apogamous (*DA*) and sexual (*DO*) gametophyte samples were used as input for STRING version 11.0 and a high threshold (0.700) was chosen for a positive interaction between a pair of proteins.

## 5. Conclusions

Gametophytes of the apomictic fern *DA* and its sexual parent *DO* share 417 proteins, out of a total of 817 identified proteins, most of which localized to the cytoplasm, chloroplasts, and the nucleus. Functions are linked to enzymatic, binding, and structural activities in the cell, and mostly associated with the response to stimuli, primary metabolism, and transcriptional regulation. A set of 145 proteins were assigned to six functional categories: growth, reproduction, phytohormone signaling and biosynthesis, gene expression, and the response to biotic and abiotic stresses. Within these functional groups, abundant protein–protein interactions could be identified. Our results provide information on the functioning and development not only of ferns, but also of plants in general. By looking at the functions of the proteins identified, possible biotechnological applications can be inferred. Specifically, homologs of *CSD1*, *GSTU19*, *SHM1*, *UBC35*, *OASA1*, *PHT3;1*, *LEA7*, *LOS4*, etc. could be transferred to plants of agronomic interest using biotechnological approaches. This may increase their ability to tolerate high or low temperatures, a lack or excess of water, high salinity, heavy metal contamination, or high UV radiation; conditions that are increasing in many parts of the world as a consequence of global climate change. Additionally, homologs of *SGT1B*, *SSI2*, *ACP4*, *OST1*, or *GLPK* could be transferred to crop plants, since they encode proteins that are involved in the response to infections caused by various microorganisms, such as bacteria and fungi, and could improve resistance to certain diseases. A better resilience towards abiotic stresses and resistance to pathogens would lead to increased crop yields that, in turn, could relieve the pressures currently being put on the environment.

## Figures and Tables

**Figure 1 ijms-23-14027-f001:**
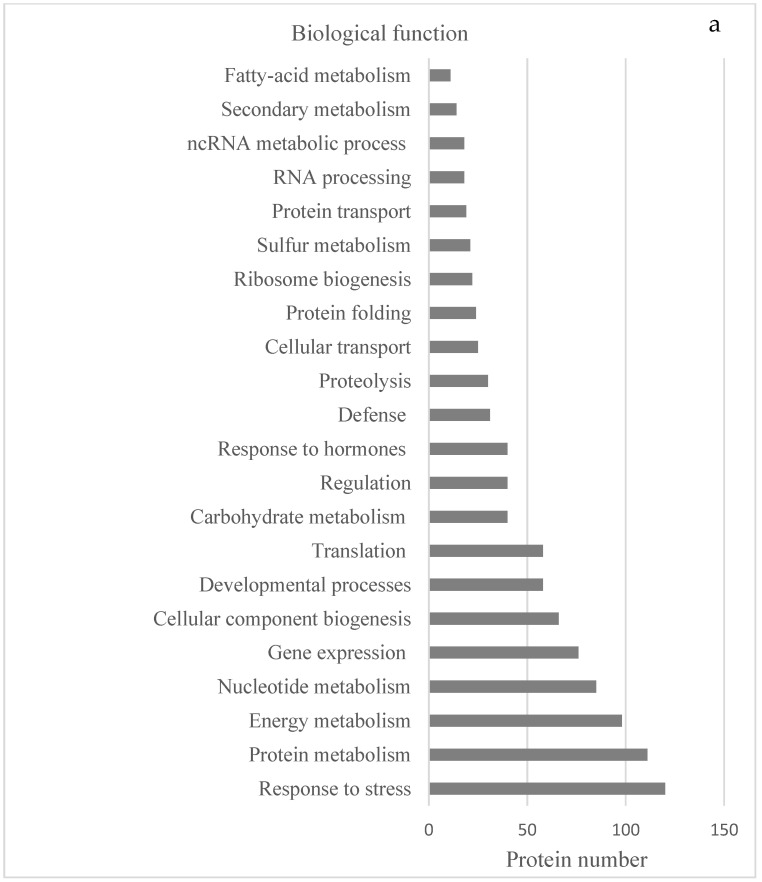
GO enrichment terms of the proteomes of *Dryopteris affinis* ssp. *affinis* and *D. oreades* according to the three main categories: (**a**) biological function, (**b**) molecular function, and (**c**) cellular component.

**Figure 2 ijms-23-14027-f002:**
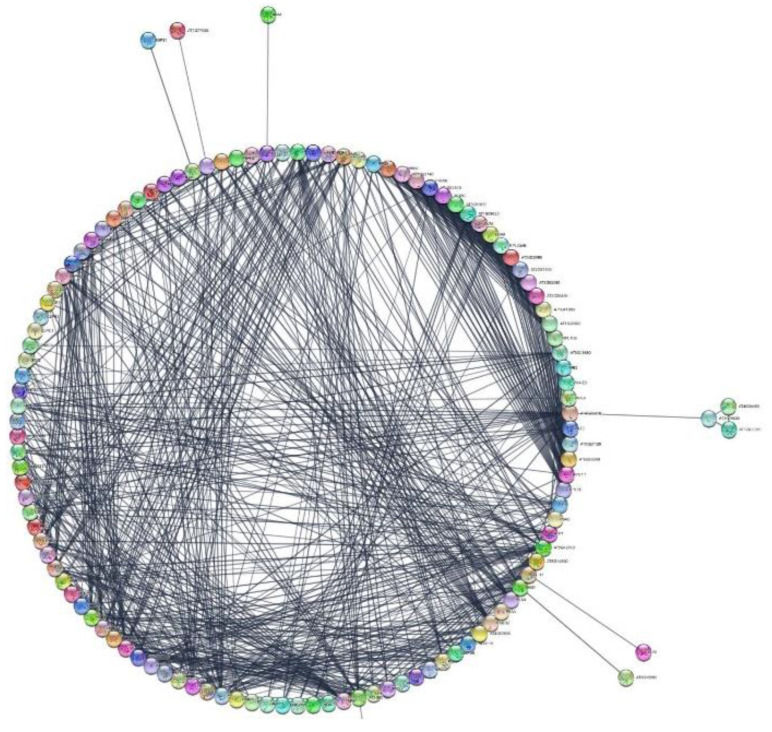
Interactome corresponding to selected proteins extracted from the gametophytes of the ferns *Dryopteris affinis* ssp. *affinis* and *D. oreades.* Data are displayed in a circular layout using Cytoscape. On the outer circle, the protein reference sets are placed, separated by color. The interactions are drawn as lines inside the circle, among proteins.

**Figure 3 ijms-23-14027-f003:**
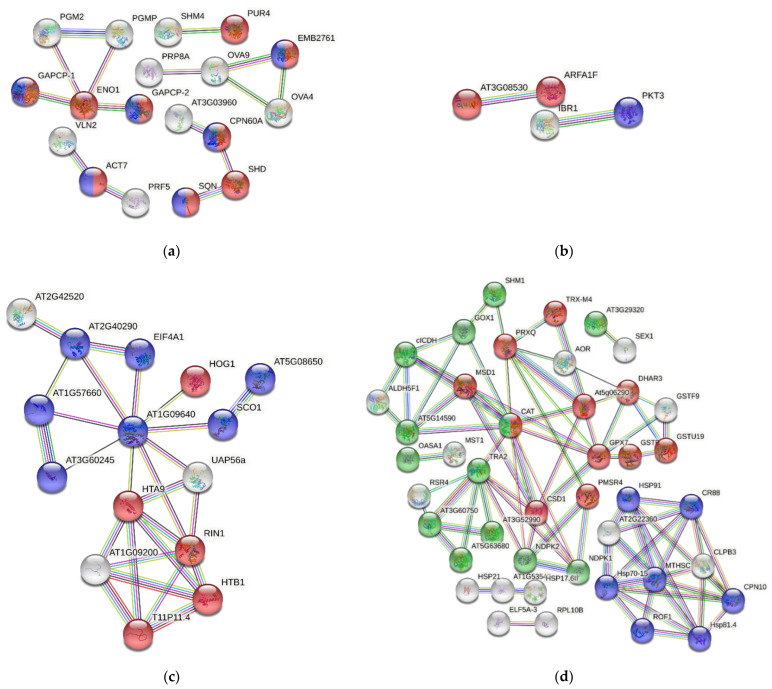
Functional STRING networks of proteins extracted from gametophytes of *Dryopteris affinis* ssp. *affinis* and *D. oreades*. Only high confidence interactions (0.7) are shown. Abbreviations of proteins are reported in Table 1 and Appendix A. (**a**) Growth and reproduction: red color refers to anatomical structure development, and blue to systems development. (**b**) Phytohormones: blue color refers to positive regulation of abscisic acid-activated signaling pathway, red to endocytosis, and green to regulation of biological quality. (**c**) Gene expression: red color refers to chromatin organization, and blue to translation. (**d**) Abiotic stress: blue color refers to protein folding, DnaJ domain and HSP90 proteins; red to glutathione metabolism, superoxide dismutase activity, and redox-active center; and green to biosynthesis of secondary metabolites.

**Figure 4 ijms-23-14027-f004:**
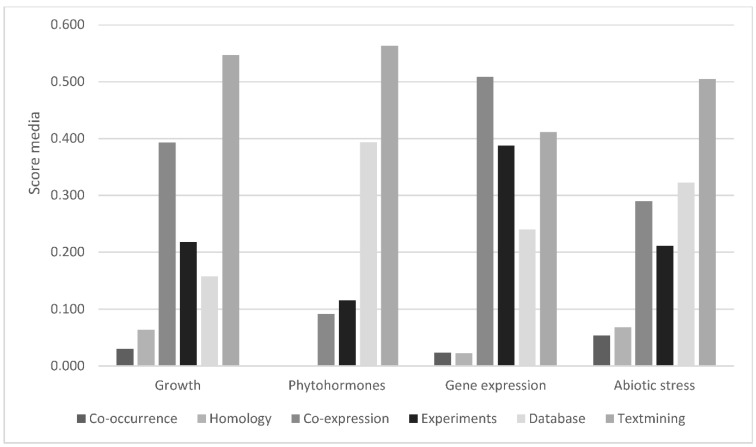
STRING functional interactions in proteins extracted from gametophytes of *D. affinis* ssp. *affinis* and *D. oreades* involved in biological functions associated with growth and reproduction, phytohormones, gene expression, and abiotic stress. Data express a modified percentage of the score values corresponding to six sources of interaction: co-occurrence, homology, co-expression, experimental data, databases, and text mining.

**Figure 5 ijms-23-14027-f005:**
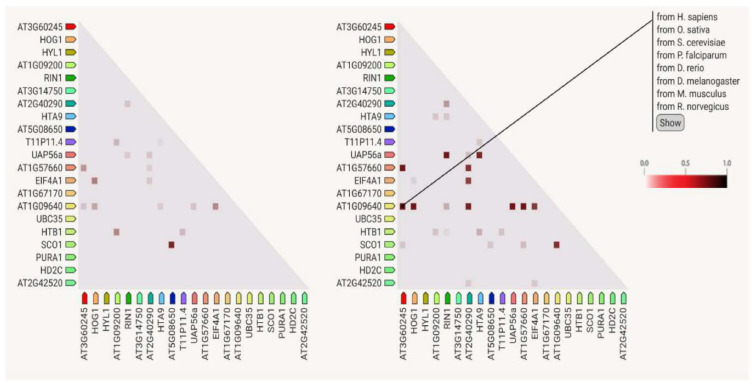
Plot of STRING co-expression interaction data among similarly regulated proteins obtained from the gametophytes of *Dryopteris affinis* ssp. *affinis* and *D. oreades*, which are associated with gene expression. Left: co-expression observed in *A. thaliana*. Right: co-expression observed in other organisms. In the triangle-matrices, the intensity of color indicates the level of confidence that two proteins are functionally connected in the organism.

**Figure 6 ijms-23-14027-f006:**
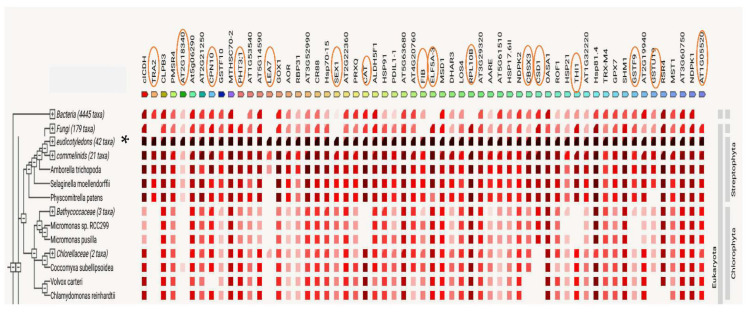
Plot of STRING co-occurrence interaction data among similarly regulated proteins obtained from the gametophytes of *Dryopteris affinis* ssp. *affinis* and *D. oreades*, which are associated with abiotic stress. In the triangle-matrices, the intensity of color indicates the level of confidence that two proteins are functionally connected in the organism. Asterisk shows the place of ferns inside Viridiplantaeae, before they split into five organization levels.

**Figure 7 ijms-23-14027-f007:**
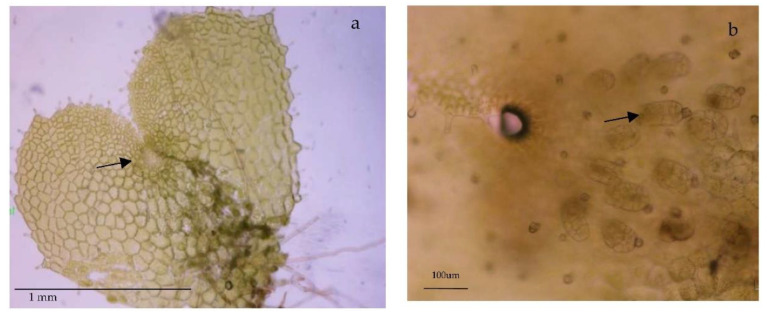
Light microscope images of mature gametophytes of: (**a**) *Dryopteris affinis* ssp. *affinis*, and (**b**) *D. oreades.* Arrows indicate an evolving apogamous center in image (**a**), or female sexual organs (archegonia) in image (**b**).

**Table 1 ijms-23-14027-t001:** Selection of proteins similarly regulated in gametophytes of *Dryopteris affinis* and *D. oreades*.

Category	Accession Number	UniProtKB/Swiss-Prot	Gene Name	Protein Name	MW (kDa)	Amino Acids	Probability
Growth	tr|A7YAU9|A7YAU9_PINTA	Q9STX5	*SHD*	Endoplasmin homolog	95	1.030	0
Growth	173702-163_1_ORF1	Q9LZF6	*CDC48A*	Cell division control protein 48 homolog E	65	894	0
Growth	149166-199_4_ORF2	A0A1P8B649	*HMA6*	Copper-transporting ATPase PAA1, chloroplastic	25	246	1.62 × 10^−22^
Growth	tr|F2DY57|F2DY57_HORVD	Q9C9C4	*ENO1*	Enolase 1, chloroplastic	51	477	0
Growth	tr|Q6AV23|Q6AV23_ORYSJ	Q56XV8	*CPN60A2*	Chaperonin 60 subunit alpha 1, chloroplastic	57	811	0
Growth	3278-1178_2_ORF2	Q9C566	*CYP40*	Peptidyl-prolyl cis-trans isomerase CYP40	40	541	0
Growth	tr|O82565|O82565_ANEPH	P53492	*ACT7*	Actin-7	42	377	0
Growth	152409-192_5_ORF1	Q42545	*FTSZ1*	Cell division protein FtsZ homolog 1, chloroplastic	47	433	2.99 × 10^−165^
Growth	tr|A9NV11|A9NV11_PICSI	Q94K05	*CCT8*	T-complex protein 1 subunit theta	59	549	0
Growth	195-2609_5_ORF2	Q41932	*PSBQ2*	Oxygen-evolving enhancer protein 3-2, chloroplastic	24	230	3.05 × 10^−77^
Reproduction	229414-114_3_ORF1	O49485	*EDA9*	D-3-phosphoglycerate dehydrogenase 1, chloroplastic	68	918	0
Reproduction	155382-187_1_ORF2	F4I6W4	*PGM2*	Phosphoglucomutase (alpha-D-glucose-1,6-bisphosphate-dependent)	68	915	0
Reproduction	40155-402_5_ORF3	Q38905	*PRF5*	Profilin-5	14	62	8.5 × 10^−59^
Reproduction	65126-313_6_ORF2	O81644	*VLN2*	Villin-2	107	1.055	0
Reproduction	88715-266_1_ORF2	Q8RXE9	*OVA4*	Tryptophan--tRNA ligase, chloroplastic/mitochondrial	45	975	0
Reproduction	12555-680_1_ORF2	Q8W4F3	*OVA9*	Glutamine--tRNA ligase, cytoplasmic	80	1.021	1 × 10^−175^
Reproduction	135880-210_3_ORF2	Q8LD94	*CYP5*	Glucan endo-1,3-beta-glucosidase, acidic isoform	25	251	0
Reproduction	152971-191_2_ORF2	Q9SIE1	*PAT*	Bifunctional aspartate aminotransferase and glutamate/aspartate-prephenate aminotransferase	55	778	0
Reproduction	116951-229_1_ORF2	F4IFC5	*EMB2761*	Threonine--tRNA ligase, chloroplastic/mitochondrial 2	45	625	0
Reproduction	76060-289_2_ORF2	Q9CAE3	*FLD*	Protein flowering locus D	52	742	0
Reproduction	tr|D7MAI5|D7MAI5_ARALL	Q9SSD2	*EMB14*	Pre-mRNA processing-splicing factor 8A	252	1.121	0
Reproduction	1422-1543_1_ORF2	O23654	*VHA-A*	V-type proton ATPase catalytic subunit A	69	623	0
Reproduction	50471-358_5_ORF2	Q9SAJ6	*GAPCP1*	Glyceraldehyde-3-phosphate dehydrogenase GAPCP1, chloroplastic	51	422	0
Reproduction	35041-428_5_ORF2	Q9M8D3	*PUR4*	Probable phosphoribosylformylglycinamidine synthase, chloroplastic/mitochondrial	156	1.407	0
Reproduction	tr|C5XKU8|C5XKU8_SORBI	Q9SRT9	*RGP1*	UDP-arabinopyranose mutase 1	40	357	0
Reproduction	186611-150_1_ORF2	Q9SIF2	*CR88*	Heat shock protein 90-5, chloroplastic	88	780	0
Reproduction	135880-210_3_ORF2	Q8LDP4	*CYP19-4*	Peptidyl-prolyl cis-trans isomerase CYP19-4	25	201	9.98 × 10^−112^
Hormones	184149-152_1_ORF2	Q9S9W2	*SDRA*	Short-chain dehydrogenase/reductase SDRA	193	254	1.35 × 10^−115^
Hormones	167580-170_3_ORF1	Q9LV09	*BOB1*	Protein BOBBER 1	20	304	4.20 × 10^−87^
Hormones	177400-159_3_ORF2	Q0WLB5	*CHC2*	Clathrin heavy chain 2	56	1.112	0
Hormones	tr|Q9ZPJ2|Q9ZPJ2_SOYBN	Q6ID97	*ARFA1F*	ADP-ribosylation factor A1F	46	153	1.95 × 10^−132^
Hormones	246374-102_1_ORF1	Q9SK82	*UGT85A1*	UDP-glycosyltransferase 85A1	14	791	1.2 × 10^−90^
Hormones	10257-745_6_ORF2	Q9FGC7	*ABA1*	Zeaxanthin epoxidase, chloroplastic	46	667	2.62 × 10^−26^
Hormones	118026-228_6_ORF2	Q9LLR6	*LTP4*	Non-specific lipid-transferprotein 4	14	56	2.04 × 10^−90^
Hormones	tr|G7JG11|G7JG11_MEDTR	Q27GK7	*TPR4*	Topless-related protein 4	88	1.001	0
Hormones	359046-40_1_ORF1	Q9FK25	*OMT1*	Flavone 3’-O-methyltransferase 1	42	562	1.75 × 10^−102^
Hormones	294468-71_2_ORF2	Q42521	*GAD1*	Glutamate descarboxylase 1	61	851	0
Hormones	53416-347_4_ORF2	A1A6K6	*AT1G76940*	Nuclear speckle RNA-binding protein A	29	233	3.86 × 10^−28^
Hormones	114827-231_2_ORF3	P13114	*CHS*	Chalcone synthase	51	395	4.93 × 10^−124^
Gene expression	118026-228_6_ORF2	A0A1P8AQD8	*HYL1*	dsRNA-binding domain-like superfamily protein	31	367	3.83 × 10^−43^
Gene expression	tr|G7JG11|G7JG11_MEDTR	Q56XG6	*UAP56A*	DEAD-box ATP-dependent RNA helicase 15	49	688	0
Gene expression	36369-420_5_ORF2	Q9LZR5	*HD2C*	Histone deacetylase HDT3	36	478	3.02 × 10^−25^
Gene expression	57659-333_6_ORF2	Q9SKZ1	*PURA1*	Transcription factor Pur-alpha 1	33	422	1.37 × 10^−90^
Gene expression	135393-211_5_ORF1	Q94A97	*UBC35*	Ubiquitin-conjugating enzyme E2 35	22	153	2.33 × 10^−107^
Gene expression	tr|A8J568|A8J568_CHLRE	Q9FMR9	*RIN1*	RuvB-like protein 1	164	1.104	0
Gene expression	239719-107_1_ORF2	O23255	*HOG1*	Adenosylhomocysteinase 1	48	485	0
Gene expression	tr|B9RUZ2|B9RUZ2_RICCO	Q9C944	*HTA9*	Histone H2A.Z variant	14	134	6.67 × 10^−76^
Biotic stress	24640-505_5_ORF2	Q940H6	*OST1*	Serine/threonine-protein kinase SRK2E	41	362	0
Biotic stress	59878-327_6_ORF2	Q9SW21	*ACP4*	Acyl carrier protein 4, chloroplastic	14	137	4.47 × 10^−29^
Biotic stress	23981-512_3_ORF1	Q38882	*PLDALPHA1*	Phospholipase D alpha 1	91	810	0
Biotic stress	9434-771_1_ORF2	Q9SUT5	*SGT1B*	Protein SGT1 homolog B	39	358	8.74 × 10^−118^
Biotic stress	30827-456_4_ORF1	P29976	*DHS1*	Phospho-2-dehydro-3-deoxyheptonate aldolase 1, chloroplastic	57	525	0
Biotic stress	tr|D7MBC8|D7MBC8_ARALL	Q9M084	*IBI1*	Aspartate-tRNA ligase 2, cytoplasmic	62	558	0
Biotic stress	3234-1184_1_ORF1	Q9M8L4	*ACP4*	Glycerol kinase	56	522	0
Biotic stress	284049-77_6_ORF2	Q94AM1	*OOP*	Organellar oligopeptidase A, chloroplastic/mitochondrial	88	791	0
Abiotic stress	177876-158_1_ORF2	Q8LE52	*DHAR3*	Glutathione S-transferase DHAR3, chloroplastic	28	258	3.08 10^−103^
Abiotic stress	10787-728_3_ORF2	P24704	*CSD1*	Superoxide dismutase[Cu-Zn] 1	15	152	2.09 × 10^−77^
Abiotic stress	180631-155_1_ORF1	O81235	*MSD1*	Superoxide dismutase[Mn] 1, mitochondrial	25	231	1.51 × 10^−104^
Abiotic stress	393073-25_4_ORF2	Q9SZJ5	*SHM1*	Serine hydroxymethyltransferase 1, mitochondrial	57	517	0
Abiotic stress	372464-34_1_ORF1	Q9LU86	*PRXQ*	Peroxiredoxine Q, chloroplastic	23	216	2.11 × 10^−94^
Abiotic stress	139597-207_6_ORF1	Q9FMU6	*PHT3;1*	Mitochondrial phosphate carrier protein 3, mitochondrial	40	375	4.08 × 10^−136^
Abiotic stress	133552-212_2_ORF2	Q96270	*LEA7*	Late embryogenesis abundant protein 7	18	169	4.76 × 10^−5^
Abiotic stress	145003-202_6_ORF2	Q93ZG7	*LOS4*	DEAD-box ATP-dependent RNA helicase 38	55	496	3.52 × 10^−177^
Abiotic stress	396305-24_6_ORF2	Q9C5R8	*2CPB*	2-Cys peroxiredoxin BAS1-like, chloroplastic	31	273	1.94 × 10^−129^
Abiotic Stress	167311-170_6_ORF2	Q9LEV3	*CBSX3*	CBS domain-containing protein CBSX3, mitochondrial	23	206	1.18 × 10^−104^
Abiotic stress	289581-74_6_ORF2	Q9ZRW8	*GSTU19*	Glutathione S-transferase U19	26	219	2.18 × 10^−54^
Abiotic stress	14237-641_3_ORF2	Q38931	*FKBP62*	Peptidyl-prolyl cis-trans isomerase FKBP62	67	551	0
Abiotic stress	14173-642_3_ORF2	P34893	*CPN10*	10 kDa chaperonin, mitochondrial	11	98	1.4 × 10^−41^
Abiotic stress	64-3566_3_ORF2	Q9LF37	*CLPB3*	Chaperone protein ClpB3, chloroplastic	114	968	0
Abiotic stress	4712-1020_6_ORF2	Q04836	*RBP31*	31 kDa ribonucleoprotein, chloroplastic	83	329	2.38 × 10^−17^
Abiotic stress	4641-1023_4_ORF2	Q38814	*THI1*	Thiamine thiazole synthase, chloroplastic	42	349	0
Abiotic stress	102649-246_6_ORF1	O64530	*STR1*	Thiosulfate/3-mercaptopyruvate sulfurtransferase 1, mitochondrial	43	379	2.01 × 10^−141^

## Data Availability

The concatenated dDB is available online at http://fgcz-r-021.uzh.ch/fasta/p1222_combo_NGS_n_Viridi_20160205.fasta (accessed on 9 November 2022).

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
