# Peer review of "The Shared Proteome of the Apomictic Fern Dryopteris affinis ssp. affinis and Its Sexual Relative Dryopteris oreades"

_ijms, 2022, doi:10.3390/ijms232214027_

Round 1

Reviewer 1 Report

A. Figure 1a = Spell correct "Defence" to "Defense" and "Nucleotid" to "Nucleotide"

B Figure 1b = The graph has 2 lines for each molecular function. Please correct.

C Line 120 = Authors mention mitochondria but there is no corresponding data in Figure 1c.

D. Table S1 and S2 were not provided for review.

E Line 574 = "At" to "For"

F Line 645 = 60min to 60 min. Please check elsewhere too.

G Line 677 = was to were

Author Response

Dear reviewer,

many thanks for your revision, and also we would like to thank the good impresion showed in the letter. Next, find the corrections done:

1. In the figure 1a, the words Defence and nucleotid were changed by "Defense" and "Nucleotide".

2 In the figure 1b the omitted lines are now shown. Thanks

3. Line 120 = The word "Mitochondria" was eliminated as it doesn´t appear in the Figure 1c. Thanks.

4. Tables S1 and S2 are provided now. Sorry for having been forgotten.

5 Line 574 = "At" to "For". Thanks. It was done.

6 Line 645 = 60min to 60 min. Thanks. It was corrected

7 Line 677 = was to were. Thanks. It was corrected.

Additionally, we have corrected some mistakes of spelling, typos, italics words...All is now highlighted in yellow. I am going also to provide the tables S1 and S2.

Hope you to find suitable the new version.

With my best wishes

Reviewer 2 Report

The manuscript titled “The Shared Proteome of the Apomictic Fern Dryopteris affinis ssp. Affinis and its Sexual Relative Dryopteris oreades (I)” reported on a proteomic study that identified 417 19 proteins shared by gametophytes of the apogamous fern Dryopteris affinis ssp. affinis and its sexual 20 relative D. oreades. This is a well-written article and I anticipate that the manuscript should be of great interest. Before recommending this article for publication, there are some shortcomings that should be resolved.

General comments

Overall, the study is well-designed and presented in a good way.

Abstract

The authors elaborated the abstract in a good way.

Introduction

This section is well written but the number of references is many. A total of 38 references are in this section and 140 all. Therefore, the authors are requested to delete the irrelevant and old references and to decrease the number of references to under 100. 

Results, Discussion, Material and Methods

These sections are written well. But the others are requested to reduce these sections as much as possible because there are many references and extra explanations.

Author Response

Dear reviewer,

I would like to thank you by the revision work done, by your diligence, and also by the good impresion showed with the ms.

Regarding the suggestions done, let me clarify:

  1. The number of references have been shorted as recomended, to 101.
  2. The repetitive paragraphs in the material and methods section, were reworded.
  3. Additionaly, the ms was revised entirely again and some spelling, typos or italics errors were corrected. All is highlighted in yellow.

Hope you to find suitable the new version.

With my best wishes